# Telephone-Based Coaching and Prompting for Physical Activity: Short- and Long-Term Findings of a Randomized Controlled Trial (Movingcall)

**DOI:** 10.3390/ijerph16142626

**Published:** 2019-07-23

**Authors:** Xenia Fischer, Jan-Niklas Kreppke, Lukas Zahner, Markus Gerber, Oliver Faude, Lars Donath

**Affiliations:** 1Department of Sport, Exercise and Health, University of Basel, 4052 Basel, Switzerland; 2Department of Intervention Research in Exercise Training, German Sport University Cologne, 50933 Köln, Germany

**Keywords:** physical activity promotion, remote, telephone coaching, SMS prompting, inactive adults

## Abstract

This study analyzed the short- and long-term efficacy of telephone coaching and short message service (SMS) prompting for physical activity (PA) promotion. Two-hundred-and-eighty-eight adults (age: 42 ± 11 years) were assigned randomly to three intervention arms: The intervention groups received 12 bi-weekly telephone calls with (coaching and SMS group) or without (coaching group) additional SMS prompts (*n* = 48 SMS). The control group received a single written PA recommendation. Self-reported and objective moderate-to-vigorous physical activity (MVPA) levels were assessed by a structured interview and by accelerometer at baseline, after the intervention (6 months), as well as after a no-contact follow-up (12 months). At post-test, self-reported MVPA increased by 173 min/week (95% CI 95 to 252) in the coaching group and by 165 min/week (95% CI 84 to 246) in the coaching and SMS group compared to control. These group differences remained similar in the follow-up test. For the objectively assessed MVPA, the coaching group increased by 32 min/week (95% CI 0.1 to 63) and the coaching and SMS group by 34 min/week (95% CI 1.6 to 66) compared to the control group. In the follow-up test, the objective MVPA levels of the intervention groups no longer differed from baseline, but group differences persisted as the control group decreased below baseline. Additional SMS prompts did not result in a further increase in PA. Telephone coaching can be considered an effective tool for PA promotion.

## 1. Introduction

Physical inactivity is considered a major public health problem [1,2]. One-third of the Swiss as well as the world’s population does not reach the recommended minimum of 150 min of moderate-to-vigorous physical activity (MVPA) per week [3,4,5]. Main barriers for physical activity (PA) in working adults are lack of time, lack of motivation, long working hours or health reasons [3,6]. Resulting inactivity is associated with a higher prevalence of non-communicable diseases including cardio-vascular diseases, type 2 diabetes, some cancers or depression [1]. Given the increasing cost of non-communicable diseases [7,8], effective PA promotion programs designed to help individuals reach a physically active lifestyle are needed.

Individually tailored interventions and programs that apply essential behavior change techniques (BCTs) were found to be effective at improving PA behavior [9,10]. Existing reviews underline the importance of self-monitoring, goal setting, action planning, social support, problem solving, and feedback on performance as key components of behavioral change [11,12,13,14,15]. On an individual level, interventions were traditionally delivered in face-to-face settings [16]. In order to reach a broader public, low-threshold interventions delivered by phone or internet have been increasingly investigated in the last two decades [10,17]. Compared to face-to-face interventions, these communication modes were found to be more cost effective [18]. Telephone coaching is particularly promising as it enables a remote but personal relation and repeated contacts to promote behavior change [10,19,20]. Short message service (SMS) based prompts require less financial and personal expenses compared to coaching and have been shown to result in small-to-moderate positive effects in preventive health behaviors [21]. Web-based interventions can be delivered to the largest number of people; however, high dropout and non-usage attrition are commonly observed when implementing such interventions [17,22]. Described intervention components showed small-to-moderate positive effects on self-reported PA levels in the short-term. The maintenance effects of PA promotion are less evident and objective measures of PA less examined. Additionally, it remains unknown which intervention components are most effective and can best be translated successfully into practice. Studies with multiple intervention arms that compare the efficacy and applicability of different intervention components are needed [23].

The present study investigated three different versions of a PA promotion program consisting of evidence based BCTs [24]. The short- and long-term efficacy of telephone coaching and SMS prompting for PA promotion were analyzed using self-reported and objectively assessed PA measures. It was built upon the hypothesis that the combination of telephone coaching and SMS prompting would result in a greater and more sustainable increase in PA compared to coaching alone and that both coaching conditions would show higher PA levels compared to a control group with minimal intervention. The present study focused on outcomes of PA (assessed at baseline, post-test, and follow-up test) as well as intervention engagement and acceptance.

## 2. Methods

### 2.1. Study Design and Participants

A three-armed randomized controlled trial with an intervention lasting six months and a follow-up period of an additional six months was conducted. The entire study was conducted without face-to-face contact with participants. A detailed description of the rationale for the study protocol and the development of the intervention can be found in a previous publication of the study protocol [24]. The randomized controlled trial was approved by the ethics committee of Northwestern and Central Switzerland (EKNZ; ID: 2016-00560) and was registered on ClinicalTrials.gov (ID: NCT02918578).

Woman and men aged 20 to 65 years achieving less than 150 min/week of MVPA [5] were eligible for the study. Self-reported PA levels achieved during one week were assessed in a screening questionnaire delivered by e-mail [25]. Participants achieving less than 150 min of PA in the screening questionnaire but more than 150 min in the baseline assessment were still included in the study. Further inclusion criteria were sufficient German language skills and a residency in Switzerland. Pregnant woman and individuals with an anticipated absence for more than three weeks during the intervention period were excluded. All participants were asked to complete the Physical Activity Readiness Questionnaire [26] to determine any potential risk factors for being physically active. Individuals with health concerns identified as a risk factor were only included in the study if their general practitioner approved their participation. Participants were recruited from August 2016 to October 2017 through flyers, newspaper advertisement, e-mail newsletters, internal advertisement, and by word-of-mouth publicity. 

The minimal sample size was estimated based on the efficacy of previous studies on PA promotion and SMS prompting for self-reported PA [10,21]. A sample size calculation for an analysis of covariance with three groups including baseline PA as a covariate was computed [27]. Results showed that 242 participants were needed for a planned statistical power of 80%, a significance level of 5%, and an expected small effect size (d = 0.21) for between group differences. Adjusting for an expected dropout rate of 15–20%, a final number of 284 participants was computed [24].

### 2.2. Intervention

The three intervention arms differed in terms of the delivery mode but were based on the same intervention content. All participants received advice on training concepts and information on how to apply BCTs in order to increase PA. Ten BCTs that were found to be effective in previous meta-analyses were communicated to all participants. This included goal setting (behavior), problem solving, action planning, review of behavioral goal(s), feedback on behavior, self-monitoring of behavior, social support, instruction on how to perform the behavior, information about health consequences, and behavior practice/rehearsal [28]. Intervention content of all study arms was individually tailored based on demographics, personal goals, and experiences [24]. In addition to the above mentioned BCTs, 25 optional BCTs (e.g., habit formation, restructuring the physical/social environment) were applied according to the participants’ needs [24]. The theoretical background of the intervention was provided by the Behavior Change Wheel Framework [29,30] and the MoVo (motivation and volition) Process Model by Fuchs et al. [31].

All participants (including the control group) had access to a password protected online platform (activity profile; http://www.movingcall.com) for planning and self-monitoring of PA [32]. Depending on the study arm, the platform was used as standalone tool or to enable interaction between participant and coach, in which case coaches had access to their participant’s profiles to engage in the designed coaching program. 

Randomization to one of the three study arms was conducted by a researcher who was not involved in the intervention. A computer-based minimization procedure stratified by age and sex was applied [33].

#### 2.2.1. Coaching Group

Participants in the coaching groups received 12 bi-weekly telephone calls. The calls had a planned duration of approximately 20 min and were conducted by the same coach over the entire intervention period. The coaching facilitated a client-centered, goal-oriented discussion between participant and coach [34]. Each coaching session contained the use of BCTs. Participants were asked to set and adapt goals, to plan their PA behavior, to analyze and overcome barriers and to gradually habituate to a physically active lifestyle. Twenty-nine exercise science and psychology students provided the coaching. Each coach completed a three-month training course, completed by a know-how-test in which knowledge of the intervention content and coaching procedure were assessed. During the intervention, coaches attended bi-weekly team meetings to elaborate on the learned content and to discuss the coaching procedure.

#### 2.2.2. Coaching and SMS Group

The coaching and SMS group received the same coaching regimen as the coaching group. Additionally, participants received four individually tailored SMS prompts during each two-week period (48 in total). Prompts were sent at varying times and there was no possibility to respond. The SMS prompts either contained information to discussed BCTs, feedback, PA knowledge or a reminder.

#### 2.2.3. Control Group (Minimal Intervention)

The minimal credible intervention of the control group consisted of a single written recommendation at the beginning of the intervention. The advice contained tailored information on how to apply BCTs in order to increase PA. An exemplary plan on weekly PA was provided within the personal profile on the online platform. Participants of the control group were asked to follow the recommendation, to adapt their personal plan, and to self-monitor their PA behavior. With the exception of the assessment periods, participants of the control group did not have contact with the study team.

### 2.3. Outcome Measures

Outcome measures were assessed at recruitment (baseline), after the intervention (post-test, after six months), and after the follow-up period (follow-up test, after twelve months). Each test period consisted of a telephone interview, objective assessment of PA by accelerometer, and online questionnaires. Additionally, participants were asked to answer a questionnaire focused on feedback and acceptance in the middle of the intervention (three months). None of the assessments required on-site presence. Interviews and data processing were conducted blinded to group allocation.

#### 2.3.1. Self-Reported Physical Activity

A structured telephone interview based on the Simple Physical Activity Questionnaire (SIMPAQ) was used to assess self-reported PA [25]. The interview assessed participants’ time spent on specific activities during the last seven days (lying, sitting, standing, other activities, walking, and sports). In addition to the original interview, participants were asked about time spent in at least moderate intensity “walking” and “other activities”. These two answers and the time spent on specific moderate-to-vigorous activities add up to the total MVPA per week. 

#### 2.3.2. Objectively Assessed Physical Activity

A wrist worn triaxial accelerometer (ActiGraph wGT3X-BT; ActiGraph, Pensacola, FL, USA) was used to objectively assess PA. ActiGraphs were mailed by post and procedures were explained by phone. Participants were instructed to wear the device for seven to ten consecutive days and nights (except for swimming or bathing), called wear time. The SIMPAQ interview was arranged at the end of the requested wear time to achieve matching assessment periods. ActiGraphs do not have a display and, accordingly, do not give any direct feedback while the participant wears it. Participants were asked to note and describe the non-wear time in a standardized document and otherwise to not let the device influence their regular behavior.

ActiGraph data were downloaded and analyzed using ActiLife version 6.13.2 (ActiGraph, Pensacola, FL, USA). ActiGraph counts were calculated using 3 axes (vector magnitude) and an epoch length of 60 s [35]. The suggested algorithm of ActiLife for wrist-worn data was not used, as it has shown to overestimate MVPA [36]. Explorative studies suggest cut off values for wrist-worn ActiGraph counts that result in similar amounts of MVPA than widely used cut off values for hip-worn ActiGraph counts [37]. According to Kamada et al. [37], accelerations with 0–1999 counts/min were coded as sedentary activities, 2000–8249 counts/min as light and accelerations, and above 8249 counts/min moderate-to-vigorous activities. Measurements were considered valid if there were at least four weekdays and one weekend day with a minimal wear time of ten hours per day [38]. Sleep time was detected using the Cole–Kripke algorithm and declared as non-wear time [39]. Wear time during the day was validated according to the algorithm of Troiano [40]. Non-wear time was then compared to participants’ notes on their non-wear-time sheet. If participants reported MVPA during their non-wear-time (e.g., swimming) these minutes of activity were added to the count-based activities and non-wear-time was adapted. Intensities of described activities were classified as moderate according to the compendium [41]. Weekly count-based MVPA was computed based on the average of all valid days per person.

#### 2.3.3. Perceived Physical Fitness

Perceived physical fitness was assessed by a 1 item proxy measure in the online questionnaire. Participants were asked to rate their current physical fitness level on a scale form one to ten [42]. 

#### 2.3.4. Intervention Fidelity and Acceptance

Participant’s adherence to the intervention was assessed in two ways. First, the completeness of the intervention delivery was categorized and evaluated. Interventions were rated as “standard”, “not standard” (incomplete intervention or interruptions of more than three weeks), “non-usage attrition” (withdrawal from intervention but participation in assessments), and “dropout” (withdrawal from study). Reasons for non-usage attrition or dropout were recorded.

Second, the engagement of participants with their activity profile was quantified by the number of active edits per months. Active edits refer to changes made within the activity profile, e.g., planning an activity on a specific day or self-monitoring (check) a completed activity.

Intervention content and completeness of delivery was documented by the coaches. Date, duration, and applied BCTs of each coaching session as well as reasons for non-executed coaching were recorded. Additionally, participants were asked about personal engagement and their perception of intervention implementation within the online questionnaire. Participants of the control group were asked how many times they read the recommendation and how they perceived the content. Participants in the coaching group were asked if the coaching was conducted regularly, how the relationship to the coach was perceived, and if the duration and frequency of calls were appropriate. Participants of the coaching and SMS group were additionally asked to rate supportiveness, as well as the content and frequency of the SMS prompts. In addition, all participants were asked questions on general satisfaction with the intervention after three months and in the post-test.

### 2.4. Data Analysis

Intention-to-treat analyses were applied to all datasets. Available data of participants completing the intervention as planned, not as planned or who withdrew participation (non-usage attrition) but completed the assessments, were analyzed in the study arm to which they were randomized. There were no data available for 13% of participants who were lost to follow-up. Participants were excluded from the analysis if they completed the assessments even though they: (a) were severely ill or injured during the entire week (5 participants in post-test and 5 in the follow-up test) and (b) if they were on sports vacation for the entire week (one participant in post-test). Unusual assessment weeks or illnesses/vacations lasting less than five full days were considered as normal variation, hence, the data were included in the analysis. 

Population characteristics are presented for each intervention arm using means and standard deviations (SD) for continuous variables, as well as counts and percentages for categorical variables. Unadjusted means and boot strapped bias-corrected, accelerated confidence intervals of self-reported and objectively assessed MVPA are shown graphically for baseline, post, and follow-up measures.

Between-group change scores (baseline to post-test and baseline to follow-up test) of self-reported and objectively assessed MVPA as well as self-reported fitness levels were assessed using linear mixed models. Group, time, and the interaction effects between groups and time were included as fixed effects. Baseline measures were added as covariates and subjects as random effects accounting for between-subject heterogeneity. Potential confounders (gender, age, BMI, education, current health complaints, how ordinary the assessed week was) were included in the models as sensitivity analysis. The models were fitted using maximum likelihood estimation. Adjusted means for within group changes and among group differences in changes from baseline to post and follow-up-test are presented with 95% confidence intervals (CIs).

Between group differences in intervention completeness were assessed using the chi-square test. Average use of the web application was compared among groups using ANOVA. All statistical analyses were computed using the STATA version 15.0 (StataCorp, College Station, TX, USA).

## 3. Results

Table 1 displays sociodemographic data of all participants at baseline. Figure 1 displays the CONSORT flow diagram of the study. Summary statistics of self-reported and count-based minutes of MVPA per week are illustrated in Figure 2.

Table 2 presents baseline adjusted changes in self-reported minutes of MVPA and differences among groups at each time point. Levels of self-reported MVPA increased in all three study arms. In the post and the follow-up tests, increases in self-reported minutes of MVPA per week were greater in the coaching and in the coaching and SMS groups compared to the control group. Means and CIs of the coaching and the coaching and SMS groups were very similar in the post and follow-up tests. Change scores of self-reported MVPA in the post and follow-up tests were similar, indicating that increases in MVPA were maintained after the end of the intervention.

Changes in objectively assessed minutes of MVPA per week and differences among groups are shown in Table 3. The average wear time of the ActiGraph was eight days (baseline: M = 8.5 d, SD = 1.3; post: M = 8.3 d, SD = 1.6; follow-up: M = 7.0 d, SD = 1.4). Objectively assessed MVPA min/week increased in the post-tests for the coaching and coaching and SMS groups but not in the control group. In the follow-up tests, the levels of MVPA in the two intervention groups no longer differed from baseline levels, but were higher than in the control group. The objectively assessed MVPA of the control group declined below baseline in the follow-up test. In the direct group comparison, change scores in objectively assessed MVPA for the two intervention groups were higher compared to the control group in the post and follow-up tests. Change scores for the coaching and the coaching and SMS groups were similar at both time points.

Perceived fitness increased in all study arms. The two intervention groups reported a greater increase in perceived fitness at both time points compared to the control group. Increases in perceived fitness were similar in the coaching and the coaching and SMS groups. Within groups, change scores at post and follow-up tests were similar. Summary statistics of perceived fitness as well as baseline adjusted change scores and among group differences can be found in Appendix A.

### Adherence, Intervention Fidelity, and Acceptance

Intervention completeness (standard, not standard, non-usage-attrition, and lost to follow-up) differed among the three study arms (Chi^2^ < 0.001). Overall seventy-seven percent of all randomized participants completed the intervention as planned. Thirteen percent of participants were classified as lost to follow-up in the post-tests and 21% in the follow-up tests. Participants who were lost to follow-up in the post-tests were also lost to follow-up in the follow-up tests. Dropout at post-test decreased from control (20%) over coaching and SMS (13%) to the coaching group (6%). Participants classified as lost to follow-up did not differ from other participants in terms of sociodemographic characteristics and baseline physical activity level. “Non-usage attrition” was higher in the coaching group (8%) compared to the coaching and SMS group (2%). Ten percent of participants in the coaching group and 9% in the coaching and SMS group were declared as “not standard” because of longer interruptions (*n* = 13) or not accomplishing all coaching sessions (*n* = 5, 3 × 11, 1 × 10, 1 × 9 coaching). Reasons for dropout and extent of intervention completeness for each intervention arm can be seen in the flow diagram (Figure 1). 

The average duration of the intervention was 22.6 weeks (SD = 1.4; range 20 to 27) for interventions rated as “standard”. Interventions rated as “not standard” lasted on average 26.8 weeks (SD = 2.6; range 24 to 32). The follow-up assessment started on average 26.7 weeks (SD = 1.7; range 22 to 36) after the end of the intervention. 

The frequency of application of the most often applied BCTs are listed in Table 4. The coaching calls lasted longer during the first (38 min, SD = 10.8) and the second (28 min, SD = 10.3) calls. The average duration of the third through to the twelfth call was 18.7 min (SD = 5.9). The use of the activity platform was higher in the coaching (median: 16.4 edits/month) and the coaching and SMS groups (19.2 edits/month) compared to the control group (2.6 edits/months). During the follow-up period, the median number of active edits per month dropped to zero in all groups. Ratings on the user friendliness of the activity profiles can be found in Appendix A.

Eighty-five percent of the participants in the control group self-reported that they read the written recommendation between one and three times. The majority (89%) considered the recommendation being comprehensive or rather comprehensive. Descriptive statistics on participants’ perception of the intervention per study arm as well as participants general acceptance can be found in Appendix A.

Participants of the coaching and coaching and SMS groups reported that the coaching calls took place regularly (93%) or rather regularly (7%). The duration (97%) and the intervals among (93%) the calls were rated as fitting. Eighty-three percent of participants rated their relationship to their coach as trusting. In the coaching and SMS group, 90% of participants reported that they always received four messages among the coaching sessions. Messages were perceived as supportive (54%) or rather supportive (32%) and the frequency was rated as fitting (83%).

In the post-test, 82% of the participants from the coaching and SMS group, 86% from the coaching group, and 19% from the control group reported that they were satisfied or rather satisfied with the intervention. Accordingly, 80% of participants in the intervention groups and 13% from the control group responded that the program did help or rather helped them to reach their personal PA-related goals.

## 4. Discussion

This randomized controlled trial examined the efficacy of telephone coaching and SMS prompting for PA promotion in adults aged 20 to 65 years. Short-term changes in PA, maintenance of PA levels as well as acceptance of the interventions were analyzed and compared to a control group receiving a minimal credible intervention.

### 4.1. Changes in Physical Activity

Telephone coaching led to higher MVPA levels in the short- and the long-term compared to a single written recommendation. Self-reported MVPA of the coaching group increased by 173 min/week at the post-test and by 112 min/week at the follow-up test compared to the control group. Similar results were achieved in the coaching and SMS group, where increases in MVPA were 165 and 113 min/week higher at post and follow-up test, respectively. The SMS prompting did not lead to a further increase in MVPA compared to coaching alone. Changes in perceived fitness were comparable to the ones observed in self-reported PA: perceived fitness increased in all study arms, although participants of the two intervention groups showed a greater increase in perceived fitness compared to the control group. All groups persistently perceived their fitness level as higher after the non-contact follow-up.

In objectively assessed MVPA, only the two intervention groups showed an increase in MVPA from baseline to post-test. The coaching group increased its MVPA by 32 min/week and the coaching and SMS group by 34 min/week compared to the control group. This increase in wrist acceleration was not maintained over time: after the no-contact follow-up, objectively assessed MVPA levels of the two intervention groups were similar to baseline MVPA. Objectively assessed MVPA of the control group remained unchanged from baseline to post-test but decreased on average by 26 min/week in the follow-up test. The two intervention groups were, therefore, similarly more active compared to the control group at the post and the follow-up tests.

Observed increases in self-reported MVPA clearly contributed to participants meeting MVPA guidelines [5]. As previous studies have documented, a 15 min increase in self-reported PA per day can lead to a 4% reduction of all-cause mortality [43]. Presented increases in self-reported PA levels as well as group differences between the control and the intervention groups can, therefore, be considered highly relevant. Observed effects of the intervention on self-reported PA can be considered sustainable as MVPA levels were maintained after the end of the intervention. However, maintenance effects of the intervention need to be interpreted cautiously, as objective PA returned to baseline.

Our findings confirm and extend existing research. Short-term increases in self-reported PA level following telephone-based PA promotion are well documented [19,20], but evidence of long-term effects are lacking [44,45]. This also applies to interventions concerning health coaching. In a recent review by Dejonghe et al. [46], only six out of the 14 included studies showed long-term efficacy. In this regard, the present study supports the findings on sustainable increases in self-reported physical activity levels. Concerning objectively assessed PA levels, there are very few studies that analyzed the short- as well as long-term effects of remote PA promotion [10,47]. One study by Van Hoecke et al. [48] compared different counseling strategies for PA promotion in older adults. Participants of an individually tailored coaching intervention as well as participants of a one-contact walking program showed larger increases in PA compared to a one-contact referral group. In accordance with the present study, self-reported PA was maintained after a no-contact follow-up, whereas objectively assessed steps (pedometer) decreased. The fact that objectively assessed PA shows smaller effects compared to self-reports has been observed in multiple studies on PA promotion. A meta-analysis conducted by Howlett et al. [15] showed small-to-moderate effects of PA intervention in healthy adults. Participants of the included studies increased their weekly activity by 31 to 247 min in the short-term and by 5 to 95 min in the long-term. However, effects on objectively assessed PA were small and, therefore, declared as non-significant in the post and follow-up assessments. In terms of the intervention content, the findings also correspond to earlier studies. As a recent review confirmed, especially those interventions applying the BCTs self-monitoring, goal setting, feedback on outcome of behavior, setting graded tasks and add objects to environment [13] as well as action planning, instruction on how to perform the behavior, prompts/cues and behavior practice/rehearsal [15] have been shown to be effective. Hence, a client-centered coaching in which autonomy is strengthened, as has been the case in the present intervention, seems to be particularly important for long-term effects [13]. Thus, presented effects of telephone coaching are consistent with the expected effects of applied BCTs.

The decrease in objectively assessed MVPA of the control group might possibly be explained by an unconscious change in PA behavior of all groups during baseline measures. Previous studies have shown higher activities of participants newly wearing a PA tracker [49]. The device applied in the present study (ActiGraph) should have had minimal influence on the behavior, as it did not allow for self-monitoring (no display). Additionally, participants were explicitly asked not to let the device influence their regular behavior. However, the mere fact of wearing a device as well as the motivation to participate in a study on PA promotion [50] might have resulted in subconscious PA level increases above usual ones.

The SMS prompting did not lead to further increases in PA compared to telephone coaching alone. Both intervention groups showed very similar PA levels in both assessment methods in the post and follow-up test. On the contrary, existing reviews show that the majority of text messaging interventions resulted in positive effects on health-related behaviors, especially when messages were individually tailored [21,51,52]. Patrick et al. [53] demonstrated that two to five text messages per day with additional monthly phone calls had a positive effect on weight loss when compared to monthly printed information. However, to our knowledge, there is no study that compared a telephone coaching intervention with or without prompts. In the present study, participants appreciated receiving SMS prompts and considered them as supportive. According to personal feedback given to coaches, participants repeatedly wished to be able to respond to the messages. Thus, the effectiveness of SMS might hypothetically have been increased by providing the option to respond. A more likely explanation for the similar results in the coaching and SMS group and the coaching group might be that the coaching alone was sufficient and that smaller beneficial effects of additional prompts were ineffective due to the ceiling effects. 

The control group showed a relevant increase in self-reported PA (86 min in the short-term and 99 min in the long-term). Participants in the control group received a minimal credible intervention which was considered comparable to tailored online information or recommendations given by a general practitioner. Participants reported to have read their recommendation one to three times and median use of the activity profile was 2.6/month. This leads to the conclusion that a minimal intervention as well as participating in a study on PA promotion [50] affected self-reported PA levels positively. Previous studies have shown bigger effects when interventions were compared to a traditional control group or a waiting list [54]. This holds true especially for long-term effects, as those are generally smaller [15]. Observed group differences in the present study, therefore, underline the positive effects of the coaching intervention.

### 4.2. Adherence and Acceptance

The dropout rate was higher in the control group compared to the coaching groups. Each participant was asked to attend the assessments independently of whether they actually interacted with the program. However, 20% of the control group did not answer the request for the post-test and reasons remain mainly unknown. Six percent of the coaching group and 13% of the coaching and SMS group were lost to follow-up. In these groups, an additional 8% (coaching) and 2% (coaching and SMS), respectively, stopped the coaching, but completed the assessments. One-third of the reasons for dropouts and non-usage attrition of the intervention groups were unrelated to the intervention (pregnancy, illness, personal issues). However, eleven participants reported lack of time and lack of motivation to continue. Observed dropout and non-usage attrition rates were low compared to other remote PA promotion interventions [10,15,17]. This allows the conclusion that the intervention was suitable for participants with limited time availability. The extent of acceptance of the program was also confirmed by the fact that 95% of participants reported being satisfied with the telephone coaching. Additional open-ended questions indicate that the personal relationship with a coach as well as the lack of any on-site presence were particularly appreciated. The duration and frequency of the calls were mainly rated as fitting. According to the feedback of coaches, it is still suggested to individualize the frequency and number of coaching sessions in a practical implementation.

The use of the online platform to plan, self-monitor, and interact with the coach was appreciated by participants. The platform was mainly used to facilitate interaction and to enrich the coaching sessions through self-monitored PA. During the follow-up period, all participants stopped using the platform. Participants were asked to keep self-monitoring and planning; however, the method used to do so was clearly communicated to be the participants’ choice. The platform remained unchanged for the entire study period. According to previous studies, it would not be surprising that usage declined if there was no new content added or updates [22].

### 4.3. Strengths and Limitations

The target group of the current study were insufficiently active adults. However, 27% of participants already fulfilled PA guidelines at baseline. These participants reported less than 150 min in a preceding screening questionnaire but were more active during the week of the first assessment. The present results are, therefore, not generalizable for a completely inactive target group. Additionally, the sample consisted of rather well-educated, employed participants with little child-caring responsibilities, who voluntarily joined the study. This self-selection recruitment further reduces the generalizability of the results.

Absolute MVPA minutes/week largely differed between objective and subjective measures: objective measures showed higher baseline activity and smaller changes from baseline to post-test compared to self-reported measures. A discrepancy between objective and subjective measures has been observed repeatedly [55] and might partially be explained by methodological limitations of both assessment methods. On the one hand, wrist acceleration does not record all forms of PA (e.g., cycling) and the amount of minutes declared as MVPA depends strongly on selected cut off values [38]. To date, there are no established cut off points for wrist-worn ActiGraph data but other devices have shown to accurately classify PA [56]. The present study applied cut off values that were considered most appropriate according to an explorative study with free-living elderly women [37]. This lack of validated cut off values for wrist-worn ActiGraph data leads to two limitations. First, the absolute minutes of MVPA need to be interpreted cautiously, as they might be inaccurate. Second, the comparability to previous studies is limited. The two limitations were accepted in order to achieve a higher wear compliance within the remote study setting [57]. Previous studies have shown high correlations of wrist- and hip-worn accelerometer data with comparable classifications of PA intensities [37,56,58]. Wrist-worn ActiGraph counts were, therefore, considered reasonable to assess between-group differences and changes over time. On the other hand, self-reported PA might be subject to recall bias and social desirability [59,60]. Increases in self-reported PA might be biased since participants focused on PA behavior during the intervention. In contrast to a study design with a traditional control group, the difference in bias between the three intervention arms might have been reduced, as the control group received a minimal credible intervention with the opportunity to use the online platform and access to a training plan example. Participants were unaware that their group served as a control group and they were likewise asked to plan, self-monitor, and increase their PA behavior. Consequently, absolute values of subjective and objective minutes of MVPA need to be interpreted cautiously. The combination of both measurement methods is still considered a strength of the study. Even though minutes of MVPA differ between the two methods, both measures clearly show benefits of remote PA counselling in the post and follow-up-test.

## 5. Conclusions

The presented three-armed study compared the efficacy of different intervention delivery modes to promote PA. The study’s no-contact follow-up enabled conclusions on the maintenance of PA behavioral changes. The combination of self-reported and objectively assessed PA tackled the limitations of both methods. In conclusion, the study shows that telephone coaching based on established BCTs leads to higher PA levels compared to a minimal intervention. Additional SMS prompts do not help to further increase PA levels. Increases in self-reported PA were maintained over time. Objectively assessed PA decreased after the end of the intervention. Nevertheless, group differences in favor of the intervention groups persisted. The coaching was highly accepted and associated with low dropout rates. Overall telephone coaching can be considered an efficacious and well-accepted tool for the promotion of a physically active lifestyle in adults of working age.

## Figures and Tables

**Figure 1 ijerph-16-02626-f001:**
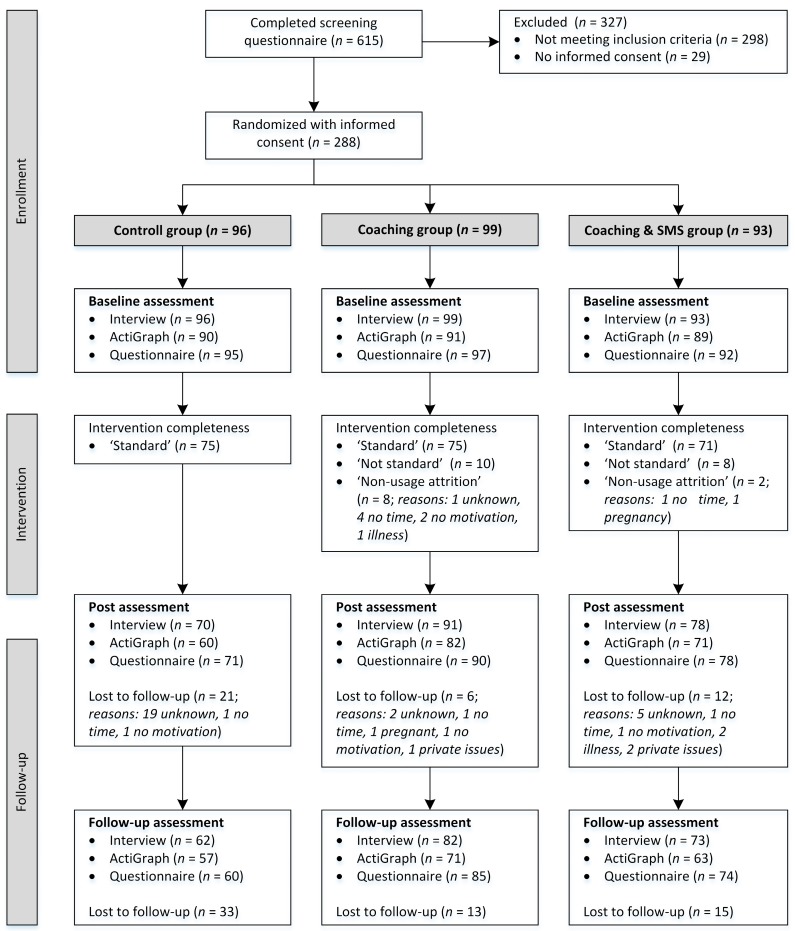
Flow diagram of the study.

**Figure 2 ijerph-16-02626-f002:**
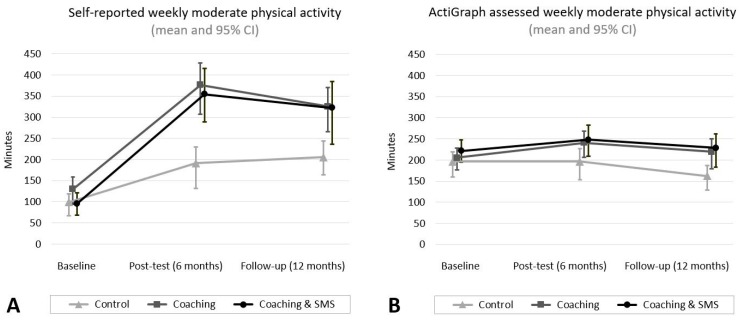
Unadjusted means and boot-strapped, bias-corrected, accelerated confidence intervals of moderate-to-vigorous physical activity (minutes/week) by intervention group. (**A**): Self-reported physical activity (baseline: *n* = 288, post-test: *n* = 239, follow-up test: *n* = 217) and (**B**): count-based physical activity (baseline: *n* = 270, post-test: *n* = 213, follow-up test: *n* = 191). SMS = Short message service.

**Table 1 ijerph-16-02626-t001:** Sociodemographic characteristic of the study population.

Variable	Control (*n* = 96)	Coaching (*n* = 99)	Coaching and SMS (*n* = 93)	Total (*n* = 288)
Age in years, mean (SD)	42.20 (11.39)	41.93 (11.12)	42.54 (11.78)	42.22 (11.39)
Age category, *n* (%)				
20–31 years	19 (19.8)	22 (22.2)	21 (22.6)	62 (21.5)
32–42 years	31 (32.3)	23 (23.2)	24 (25.8)	78 (27.1)
43–53 years	27 (28.2)	39 (39.4)	32 (34.4)	98 (34.0)
54–65 years	19 (19.8)	15 (15.2)	16 (17.2)	50 (17.4)
Gender, *n* (%)				
Female	64 (66.7)	69 (69.7)	64 (68.8)	197 (68.4)
Male	32 (33.3)	30 (30.3)	29 (31.2)	91 (31.6)
BMI in kg/m^2^, mean (SD)	26.43 (5.33)	25.26 (4.28)	26.24 (4.99)	25.97 (4.89)
BMI Category, *n* (%)				
Underweight (<18.50)	2 (2.1)	2 (2.0)	1 (1.1)	5 (1.7)
Normal weight (18.50–24.99)	44 (45.8)	54 (54.6)	43 (46.2)	141 (49.0)
Overweight (25.00–29.99)	28 (29.2)	27 (27.3)	32 (34.4)	87 (30.2)
Obese (≥30.00)	22 (22.9)	16 (16.2)	17 (18.3)	55 (19.1)
Occupation, *n* (%)				
Employed	83 (86.5)	84 (84.9)	76 (81.7)	243 (84.4)
Student	4 (4.2)	8 (8.1)	8 (8.6)	20 (6.9)
House wife/husband	4 (4.2)	2 (2.0)	5 (5.4)	11 (3.8)
Pensioner	2 (2.1)	3 (3.0)	1 (1.1)	6 (2.1)
Unemployed	1 (1.0)	-	3 (3.2)	4 (1.4)
No response	2 (2.1)	2 (2.0)	-	4 (1.4)
Highest education level, *n* (%)				
Compulsory education	1 (1.0)	1 (1.0)	2 (2.2)	4 (1.4)
Apprenticeship	28 (29.2)	30 (30.3)	27 (29.0)	85 (29.5)
High school	21 (21.9)	21 (21.2)	22 (23.7)	64 (22.2)
University	39 (40.6)	39 (39.4)	37 (39.8)	115 (39.9)
Doctorate	5 (5.2)	7 (7.1)	5 (5.4)	17 (5.9)
No response	2 (2.1)	1 (1.0)	-	3 (1.0)
Yearly household income, *n* (%)				
<50,000 CHF	13 (13.5)	19 (19.2)	11 (11.8)	43 (14.9)
50,000–100,000 CHF	40 (41.7)	42 (42.4)	47 (50.5)	129 (44.8)
>100,000 CHF	41 (42.7)	35 (35.4)	32 (34.4)	108 (37.5)
No response	2 (2.1)	3 (3.0)	3 (3.23)	8 (2.8)
Family status: Number of children bellow 18 years, *n* (%)				
No children	69 (71.9)	63 (63.6)	63 (67.7)	195 (67.7)
1 child	9 (9.4)	8 (8.1)	9 (9.7)	26 (9.0)
2 children	10 (10.4)	19 (19.2)	12 (12.9)	41 (14.2)
3–4 children	3 (3.1)	3 (3.0)	1 (1.1)	7 (2.4)
Missing response	5 (5.2)	6 (6.1)	8 (8.6)	19 (6.6)

**Table 2 ijerph-16-02626-t002:** Adjusted changes in self-reported minutes MVPA (moderate-to-vigorous physical activity) within groups and differences among groups at each time point.

M	Adjusted Mean Change from Baseline in min/week (95% CI)			Pairwise Comparison: Differences among Groups in Change from Baseline (95% CI)		
	Control	Coaching	Coaching and SMS	Coaching versus Control	Coaching and SMS versus Control	Coaching and SMS versus Coaching
6	86.9 (28.1 to 145.7)	259.9 (208.1 to 311.7)	252.3 (196.5 to 308.1)	173.0 (94.5 to 251.5)	165.4 (84.4 to 246.3)	−7.6 (−83.9 to 68.7)
12	98.9 (36.8 to 161.1)	211.4 (157.0 to 265.7)	212.1 (154.6 to 269.6)	112.4 (29.7 to 195.2)	113.2 (28.6 to 197.8)	40.9 (−37.2 to 119.0)

M = Months.

**Table 3 ijerph-16-02626-t003:** Adjusted changes in count-based minutes of MVPA within groups and differences among groups at each time point.

M	Adjusted Mean Change from Baseline in min/week (95% CI)			Pairwise Comparison: Differences among Groups in Change from Baseline (95% CI)		
	Control	Coaching	Coaching and SMS	Coaching vs. Control	Coaching and SMS vs. Control	Coaching and SMS vs. Coaching
6	−5.1 (−28.7 to 18.6)	26.5 (5.8 to 47.1)	28.5 (7.0 to 50.0)	31.5 (0.1 to 62.9)	33.5 (1.6 to 65.5)	2.0 (−27.8 to 31.8)
12	−26.1 (−50.1 to −2.1)	6.9 (−14.7 to 28.5)	15.6 (−6.8 to 38.0)	33.0 (0.7 to 65.2)	41.7 (8.9 to 74.5)	8.7 (−22.4 to 39.8)

M = Months.

**Table 4 ijerph-16-02626-t004:** Frequency of application of used behavior change techniques.

Behavior Change Technique	Mean	SD
Action planning	7.2	3.2
Feedback on behavior	7	3
Self-monitoring of behavior	6.3	3.1
Problem solving	5.2	2.4
Goal setting (behavior)	4.7	2.6
Review of behavioral goal (s)	4.5	2.8
Instruction on how to perform the behavior	3	2.4
Social support	2.9	2.4
Habit formation	2.9	2.6
Information about health consequences	2.1	1.6
Goal setting (outcome)	1.6	2
Behavior practice/rehearsal	1.3	1.5

Mean and standard deviation (SD) of frequency of application per person over the 12 coaching sessions.

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
