# Peer review of "Telephone-Based Coaching and Prompting for Physical Activity: Short- and Long-Term Findings of a Randomized Controlled Trial (Movingcall)"

_ijerph, 2019, doi:10.3390/ijerph16142626_

Round 1

Reviewer 1 Report

Very well written and well constructed randomized controlled trial. The program and the statistical planning are well articulated. The paper reads a bit long and the authors may want to consider shortening some of the discussion regarding the use of the accelerometers and focus more on the program itself.

I do not have a lot of feedback but there are some comments below:

Line 43: Consider rewording “individually tailored procedure”

Line 49: Consider rewording “ during the last years”

Line 63: Describe the behavior change theory you based your program on

Line 107: How much do the researchers think that the use of the online platform affected results? Would utilizing phone calls without this platform have had the same intended effect? I wonder if this needs to be stated in the introduction

Line 112: Is this supposed to say randomization?

Given that you have both subjective and objective PA records it would be useful to see some analyses comparing these so that you could direct other researchers as to whether use of a subjective PA record is adequate to show effectiveness in these types of studies or whether objective is necessary. This is mentioned in the discussion but statistical analyses would be useful.

I think more needs to be written in the discussion about the lack of continued physical activity into the follow up period. What have other studies shown? This is a big factor in physical activity planning as it is not useful as a program if it is not continued.

Author Response

You can find our answer in the attached PDF, thank you very much.

Reviewer 2 Report

Dear authors 

I read the paper entitled "Telephone-based Coaching and Prompting for Physical Activity: Short and Long-term Findings of a Randomized Controlled Trial (Movingcall)". It's an interesting manuscript on the fight against a sedentary lifestyle, with the topic on the usefull of phone call with/without SMS message. 

The sample size is good with an appropriate study design.

However the review suggest to implementing the statistical analysis:

it's not clear the statistical power of PA level in the 3 arm in the 3 different time point, a descriptive analysis of the differences is not sufficient for the purpose of the study.

In addition in discussion section maybe the underline the duration of objective measure with Actigraph (8 days) should help the authors to justify the differences in comparison with questionnaire in PA results.

Author Response

(The authors gave the same response as above.)
